# IL18 Receptor Signaling Inhibits Intratumoral CD8^+^ T-Cell Migration in a Murine Pancreatic Cancer Model

**DOI:** 10.3390/cells12030456

**Published:** 2023-01-31

**Authors:** Elena Nasiri, Malte Student, Katrin Roth, Nadya Siti Utami, Magdalena Huber, Malte Buchholz, Thomas M. Gress, Christian Bauer

**Affiliations:** 1Department of Gastroenterology, Endocrinology, Infectious Diseases and Metabolism, University Hospital Marburg, Philipps University Marburg, 35043 Marburg, Germany; 2Department of Internal Medicine I, University Hospital Ulm, 89081 Ulm, Germany; 3Core Facility Cellular Imaging, Center for Tumor Biology and Immunology, Philipps University Marburg, 35043 Marburg, Germany; 4Institute for Medical Microbiology and Hospital Hygiene, Philipps University Marburg, 35043 Marburg, Germany

**Keywords:** live cell imaging, intravital two-photon microscopy animal models for intravital imaging, T-cell migration, tumor spheroid coculture model, pausing phases/arrest coefficient, fluorophore, cytokines, T-cell plasticity, pancreatic carcinoma

## Abstract

In pancreatic ductal adenocarcinoma (PDAC), the infiltration of CD8^+^ cytotoxic T cells (CTLs) is an important factor in determining prognosis. The migration pattern and interaction behavior of intratumoral CTLs are pivotal to tumor rejection. NLRP3-dependent proinflammatory cytokines IL-1β and IL-18 play a prominent role for CTL induction and differentiation. Here, we investigate the effects of T-cellular IL-1R and IL-18R signaling for intratumoral T-cell motility. Murine adenocarcinoma cell line Panc02 was stably transfected with ovalbumin (OVA) and fluorophore H2B-Cerulean to generate PancOVA H2B-Cerulean tumor cells. Dorsal skinfold chambers (DSFC) were installed on wild-type mice, and PancOVA H2B-Cerulean tumor cells were implanted into the chambers. PancOVA spheroids were formed using the Corning^®^ Matrigel^®^-based 3D cell culture technique. CTLs were generated from OT-1 mice, *Il1r^−/−^* OT-1 mice, or *Il18r^−/−^* OT-1 mice and were marked with fluorophores. This was followed by the adoptive transfer of CTLs into tumor-bearing mice or the application into tumor spheroids. After visualization with multiphoton microscopy (MPM), Imaris software was used to perform T-cell tracking. Imaris analysis indicates a significantly higher accumulation of *Il18r^−/−^* CTLs in PancOVA tumors and a significant reduction in tumor volume compared to wild-type CTLs. *Il18r^−/−^* CTLs covered a longer distance (track displacement length) in comparison to wild-type (WT) CTLs, and had a higher average speed (mean track speed). The analysis of instantaneous velocity suggests a higher percentage of arrested tracks (arrests: <4 μm/min) for *Il18r^−/−^* CTLs. Our data indicate the contribution of IL-18R signaling to T-cell effector strength, warranting further investigation on phenomena such as intratumoral T-cell exhaustion.

## 1. Introduction

Pancreatic ductal adenocarcinoma (PDAC) is the fourth leading cause of cancer death in the Western world [1]. Despite refined strategies to treat advanced pancreatic carcinoma with combination chemotherapies such as oxaliplatin, irinotecan, fluorouracil, and leucovorin (FOLFIRINOX), prognosis is dismal. The aggressive tumor biology of PDAC is characterized by extensive local tumor infiltration and early metastasis. A strong desmoplastic reaction is thought to limit the accessibility of cytotoxic drugs to cancer cells [2,3]. Earlier reports proposed that the inhibition of prostromal signaling cascades leads to an increase in the efficacy of chemotherapeutic drugs by increasing drug delivery [4]. However, clinical trials following up on this stromal depletion hypothesis were unsuccessful and did not reproduce obtained results in genetically engineered mouse models [5]. Furthermore, stromal elements restrained rather than supported PDAC tumor progression. Within this highly complex web of cellular interactions, CD8^+^ cytotoxic T cells (CTLs) stand out as pivotal for the rejection of malignant cells on the basis of the expression of antigenic material that is recognized by the T-cell receptor in the context of MHC-I bound antigen-derived peptides. Very much like the role of the stroma in pancreatic cancer, beliefs about the role of cytotoxic T cells and immunotherapy in pancreatic carcinoma had to be re-evaluated and refined. Traditionally, pancreatic cancer was believed to be highly inert to tumor immunotherapy [2], characterized by low antigenicity and low immunogenicity. This was thought to result in little to no antigenic burden that could be exploited for immunotherapy, and very hard conditions for those few T cells to infiltrate the tumors. In general, reports on immunotherapeutic approaches in pancreatic carcinoma were characterized by low patient numbers and rather anecdotal findings of therapeutic success [6]. However, more recently, certain subgroups of pancreatic cancer were responsive towards checkpoint inhibitor therapy [7]. Mismatch repair deficiency, which results in mutagenic events leading to neoantigens and a high antigenic burden, predicted the response of solid tumors to PD-1 blockade. This indicated that CD8^+^ CTLs could successfully eradicate pancreatic carcinoma tumor cells. Admittedly, less than 2% of PDAC are mismatch-repair-deficient. However, genomic profiling has allowed for the identification of pancreatic cancer subgroups that might be more susceptible to immunotherapy [8]. Recently, proteogenomic approaches have identified PDAC entities that are characterized by immune exclusion [9], which, in turn, might guide therapeutic decisions in the future.

All this indicates that CD8^+^ T-cell effector function is pivotal to the immunological eradication of cancer cells from the organism. The interaction of CD8^+^ T cells with their target cells is characterized by the formation of an immunological synapse with the release of perforin and granzyme, and the production of effector cytokines TNF and IFN-γ [10]. As cellular contacts are necessary to form MHC-I-mediated contacts with target cells that express the cognate antigen, T-cell motility determines the frequency of cytotoxic interactions. T-cell migration patterns within the tumor stroma are characterized by low velocity when compared to migration within lymph nodes, which is around 25 μm/min [11]. Intratumoral migration is undirected, described as a “random walk” pattern [11], and characterized by abrupt changes in speed and angle of movement direction [12,13,14,15].

T-cell motility is a dynamic, multistep process characterized by the reorganization of the cytoskeleton, resulting in the polarization of cellular morphology [16]. As an initial response of, e.g., a cytotoxic CD8^+^ T cell towards migration-promoting stimuli, the T cell extends protrusions in the direction of migration. The leading edge contains networks of filamentous actin that enable the cell to dynamically probe the environment. This leading edge is particularly sensitive to the engagement of Fc and T-cell antigen and chemokine receptors [17]. CD8^+^ cytotoxic T cells use LFA-1-ICAM interactions to crawl into junctions between target cells [18]. Multiphoton microscopy demonstrated that CTLs reach their target site by moving through an adjacent stroma [19]. This seems to be guided by chemokines such as CXCL12, CXCL10 and CCL2, produced by various cell types such as tumor cells, and tumor-associated macrophages and fibroblasts [20,21]. Upon the recognition of their high-affinity cognate antigen, the fast-paced, scanning pattern of undirected migration is converted into a slower-velocity motility pattern [13,22]. Here, CTLs demonstrate migration arrest phases resulting from prolonged engagement with individual target cells. During those arrest phases, the engagement of adhesion-promoting and costimulatory molecules supports and enhances release of cytotoxic granules. Therefore, T-cell migration patterns indicate the functional state of effector T cells, and alterations in this migration pattern might be associated with T-cellular dysfunction.

The phenomenon of intratumoral T-cell dysfunction bears certain similarities to T-cell exhaustion, which was first described in models of viral infection [23]. T-cell exhaustion is induced by the persistent activation of the T-cell receptor, and characterized by the expression of coinhibitory receptors, such as PD-1 and TIM-3 [24], resulting in a loss of effector function. Recently, our group found that IL-18 receptor signaling in intratumoral CD8^+^ T cells promotes impaired tumor rejection, the increased expression of coinhibitory receptors, and the loss of cytotoxic cytokine production in a murine subcutaneous PancOVA and orthotopic KPC tumor model (Lutz et al., Cancer Immunology Research, *in press*). IL-1 receptor signaling also promoted T-cell exhaustion, indicating the general involvement of the IL-1 receptor family in T-cell exhaustion.

The causal relationship between T-cell functional status and T-cell migration is not well-described. On the basis of the finding that IL-1/18R signaling influences T-cell plasticity, we hypothesized that signaling through these receptors influences the migration and interaction pattern of T cells. T-cell migration is necessary for interacting with the antigen-presenting cells and target cells, as T cells survey the tissues for their cognate antigens. The dynamics of T-cell migration are governed by both environmental and intrinsic mechanisms, and are indicative of T-cell effector function. Here, we demonstrate that the intrinsic T-cell signaling of Nlrp3-dependent cytokines IL-18 and IL-1β influences intratumoral T-cell migration patterns, which is indicative of alterations to cytotoxic effector function.

## 2. Materials and Methods

### 2.1. Cell Lines

Murine Panc02 cells that had been transfected with ovalbumin (PancOVA) [25] were used for the tumor and spheroid models. To visualize the tumor cells with MPM, the plasmid pCS H2B Cerulean was stably transfected into Panc02 and PancOVA using a lentiviral transfection system. Tumor cells were cultivated in DMEM media containing 10% FCS and 2% P/S in T75 flasks (Sarstedt, Nümbrecht, Germany). For the positive selection of PancOVA, 500 mg/l G418 was added to complete the DMEM media.

### 2.2. Mice

All animal experiments were approved by the regional agency of animal experimentation (Regiungspräsidium Gießen, Aktenzeichen G78/2016). C57BL6/J mice were purchased from Jackson Laboratories. *Il18r^−/−^* (B6.129P2-Il18r1tm1Aki/J) and *Il1r^−/−^* (B6.129S7-Il1r1tm1Imx/J) mice, all against a C57BL6/J background, were mated with OT-1 TCR transgenic mice. Mice were housed under specific pathogen-free (SPF) conditions in the animal facility at Philipps University Marburg. All mice used for the experiments were male and aged between 2 and 5 months. Recipient mice used for the DSFC experiments weighed at least 21 g.

### 2.3. Murine T-Cell Isolation and In Vitro Differentiation

WT, *Il1r^−/−^* or *Il18r^−/−^* OT-1 mice were used for T-cell isolation. Mice were sacrificed by cervical dislocation. Spleen and lymph nodes were meshed through a 30 μm strainer (Miltenyi, Gladbach, Germany). After centrifugation and erythrocyte lysis, the cells were adjusted to 4 × 10^7^ cells/mL. Then, 5 μM of the OVA peptide SIINFEKL (Invitrogen, Waltham, MA, USA) was added to the naïve T cells and incubated at 37 °C for 1 hour. T cells were seeded with T-cell media (RPMI-1640 medium supplemented with *10%* FCS, 1% Pen-Strep, 0.1% β-mercaptoethanol, and 1% L-glutamine) and 10 ng/mL IL 12 (Peprotech, Cranbury, NJ, USA) in T75 flasks. After 48 h, the medium was changed, and 20 ng/mL IL-2 (Peprotech, Cranbury, NJ, USA) was added. This was followed by another 48 h of incubation with a subsequent medium change and IL-2 administration. After 5 days of incubation, the effector cells were used for further experiments.

### 2.4. Staining of Cytotoxic T Cells

CTLs were transferred from T75 flasks into 15 mL Falcon tubes, and the cell number was adjusted to either 1 × 10^5^ cytotoxic T cells (spheroid coculture) or 1 × 10^6^ CTLs (DSFC) and stained for 45 min at 37 °C with 1 µM CellTracker™ Orange CMTMR (Invitrogen, stock solution 1 mM) in PBS. CellTracker™ Orange CMTMR is nontoxic and, after passing through a cell’s outer cell membrane, is converted into a reaction product that cannot penetrate the cell membrane. CellTracker dyes transfer to daughter cells, but do not transfer to adjacent cells. After staining, CTLs were resuspended in T-cell media and added to spheroids or resuspended in 50 µL PBS for intraperitoneal injection into tumor-bearing mice with DSFC.

### 2.5. DSFC

Dorsal skinfold chambers (DSFCs) were attached on mice in order to perform intravital MPM on intratumoral T cells, as described before [12]. For that lateral, titanium clamps (APJ Trading Co, Inc., Ventura, CA, USA) were attached to a skinfold on the back of mice. One side of the skin was removed, and a coverslip was inserted. A total of 1 × 10^6^ Panc02 H2B Cerulean/PancOVA H2B Cerulean tumor cells were injected 1 day later s.c. into the imaging chamber under the cover slip created via the removal of the skin flap. Seven days later, 1 × 10^6^ WT, *Il1r^−/−^*, or *Il18r^−/−^* OT-1 CTLs were adoptively transferred into the tumor-bearing mice after T-cell staining.

### 2.6. Spheroid Generation and Coculture

Spheroids generated with the Corning^®^ Matrigel are three-dimensional structures that demonstrate both morphological and cellular similarities to in vivo tumors, as Matrigel mimics the extracellular matrix. In order to generate Panc02/PancOVA spheroids, Panc02 or PancOVA cells were adjusted to a cell number of 1 × 10^5^ in 25 μL complete DMEM media (+10% FCS + 2% P/S), mixed with 25 µL Corning^®^ Matrigel, and pipetted onto an Ibidi^®^ 35 mm µDish. After the cell–Matrigel drop had solidified, 2 mL of complete DMEM media was added. Spheroids formed within 48 h in the incubator. After 48 h, WT, *Il1r^−/−^*, or *Il18r^−/−^* CTLs were carefully added to the spheroids at the edge of the dish. Microscopic in vivo imaging started 4 hours after coculture for the early time points, and 18 h after coculture for the late time points.

### 2.7. Intravital Multiphoton Microscopy

Intravital microscopy was performed with a multiphoton microscope from Olympus (FVMPE-RS, Tokyo, Japan) with two pulsed lasers (Spectra Physics: Insight DeepSee 690–1300 nm, MaiTai Ti:Sa 690–1040 nm) and a water immersion objective (25×/NA 1.05). The Insight DeepSee was set to 1020 nm for the CellTracker™ Orange-labeled CTLs (545–620 nm). The MaiTai Ti:Sa laser was set to a wavelength of 835 nm for H2B Cerulean (460–500 nm). Anesthetized mice (with 1.5–2% isoflurane via a mask, controlled with a pulse oximeter) or 35 mm Ibidi^®^ dishes with spheroids and CTLs were placed to the heated stage/insert. Images were captured with 640 × 640 pixels (509 µm), a total layer thickness of 60 µm, and a step size of 5 µm. The recording time was 30 min with a time interval of 30 s.

### 2.8. Analysis

First, Fiji (ImageJ, MD, USA) software was used to subtract the Cerulean signal from the Celltracker™ Orange signal. Then, analysis continued with Imaris ×64 8.3.1 and 9.7.2 (Bitplane, Oxford, UK). For data from DSFC, the surface of the pancreatic tumor was manually determined, as the collagen fibers of the tumor environment were also detected with the cyan filter via second harmonic generation. For spheroids, the surface algorithm of Imaris with a high smooth factor was chosen in order to achieve a surface covering the whole spheroid. For both surfaces (tumor, spheroid), a drift correction was implemented when necessary. A mask was set to discriminate between CTLs inside the surface (infiltrating) and those that were outside the surface. Additionally, outside CTLs were split into cells closer than 10 µm to the surface (approaching) and those that were further away (peripheral). For CTLs, the spot algorithm of Imaris was chosen. From Imaris, the volume of spheroids, the number of infiltrating, approaching, and peripheral CTLs, the average speed, track displacement, and instantaneous velocity were exported as Excel files. Data are presented graphically using Prism 8 and 9 (GraphPad Prism, San Diego, CA, USA). With the R Studio Team program and its software packages tidyverse, dplyr, dslabs, and plotly, the stopping phases (Pausing Phases) and arrests (Arrests) could be calculated with the data of instantaneous velocity. An arrest was defined as a track segment characterized by average velocity below 4 μm/min for ≥30 s. The arrest coefficient is defined as the fraction of time during which each individual cell is immobilized, calculated between 30 s intervals and a 4 μm/min threshold.

### 2.9. RNA-Seq Analysis and Bioinformatics

RNA-Seq data that were initially reported in a manuscript from our group currently in press (Lutz et al., IL-18 receptor signaling regulates tumor-reactive CD8^+^ T-cell exhaustion via the activation of the IL-2/STAT5/mTOR pathway in a pancreatic cancer model, Cancer Immunology Research, *in press*) were reanalyzed with a focus on the expression of motility-associated genes. The sequencing data that were used are available under accession number GSE200248. In short, 3 days after the adoptive transfer of WT versus *Il18r^−/−^* OT-1 CD8^+^ CTLs into CD45.1^+^ mice bearing subcutaneous PancOVA tumors (Day 7 after inoculation at adoptive transfer), CD45.2^+^ CD8^+^ T cells were sorted on a FACS Aria III. RNA was purified using RNeasy Plus Micro Kit (Qiagen, Hilden, Germany). For the RNA-Seq of the transferred CTLs, 1 ng of total RNA was used for cDNA synthesis and amplification with Takarabio SmartSeqv4 kit. cDNA (1 ng) was used for library preparation using the Nextera XT DNA Sample Prep Kit (Illumina, San Diego, CA, USA). Barcoded RNA-Seq libraries were sequenced (150 PE) at Novogene (Cambridge, UK) on an Illumina NovaSeq, as described before (Lutz et al., Cancer Immunology Research, *in press*).

### 2.10. Pathway and Process Enrichment Analysis

To identify the biological pathways that are significantly altered in *Il18r^−/−^* CTLs in comparison to WT CTLs, pathway enrichment analysis was performed. An open-source, free gene annotation and analysis resource, Metascape, was used to characterize the RNA-Seq results [26]. Enrichment analysis was used to define up- and downregulated genes within the RNA-Seq dataset (log2FC 0.75, padj < 0.01). Different ontology sources, namely, GO Biological Processes, KEGG Pathway, Reactome Gene Sets, CORUM, Wiki Pathways, and PANTHER Pathway, were used. Functional enrichment GO analysis was performed via Metascape. Using Cytoscape, the top 20 involved pathways were used to prepare a network plot showing 20 GO biological terms. Each node represents a term, and the node size is proportional to the number of input proteins grouped in each node. Nodes belonging to the same cluster are presented with the same color. Each edge connects the terms with similar score (>0.3).

### 2.11. Statistics

The Kolmogorov–Smirnov test for unpaired samples of nonparametric distribution was used to calculate the statistical significance of the differences between individual groups. The statistical significance of a few values in a group was calculated using the two-tailed *t*-test. Results are presented as the mean with standard deviation; statistical significances with *p* values < 0.05 are shown with *.

## 3. Results

### 3.1. IL-18R Deficiency Enhances Intratumoral CTL Migration

In order to visualize the infiltration of CD8^+^ CTLs into pancreatic cancer tissue, intravital multiphoton imaging was used. For this, IL-18R1- and IL-1R-deficient, and WT mice of the OT-1 strain, in which CD8^+^ T cells are specific for a peptide epitope of model antigen ovalbumin (OVA257-264), were generated in vitro by stimulating T cells with their cognate antigen in the presence of IL-2 and IL-12 (*Il18r^−/−^*, *Il1r^−/−^*, WT; Figure 1A). Murine pancreatic adenocarcinoma cell line Panc02, derived from a 3-methylcholanthrene-induced tumor in a C57Bl/6 mouse [27], was stably transfected with ovalbumin (PancOVA), as described before [25]. PancOVA cells were stably transfected with fluorochrome H2B-Cerulean. In order to visualize H2B-Cerulean-labelled cancer cells and transferred fluorophore-labelled T cells, mice were implanted with dorsal skinfold chambers (DSFCs). Through the removal of a skin flap, an imaging chamber is created in which inoculated PancOVA cells can be visualized. CTLs were adoptively transferred intraperitoneally (i.p.) into mice 7 days after the injection of PancOVA cells. The multiphoton imaging of CTLs through DSFCs was performed 36 h after transfer; the analysis of imaging data was conducted in Imaris (Appendix A). Depending on the distance between individual T cells and PancOVA cells, CTLs were divided into infiltrating T cells (within the tumor margins), approaching T cells (<10 µm distance to tumor surface), and peripheral T cells (>10 µm distance to tumor surface) (Figure 1B). Less than 20% of the WT CTLs in representative image fields infiltrated Panc02 tumors, with more than half of the T cells remaining in the periphery (Figure 1C). This was in stark contrast to PancOVA tumors, where WT, *Il1r^−/−^*, and *Il18r^−/−^* CTLs showed significantly higher infiltration rates of around 40%. The higher infiltration of the tumor parenchyma by *Il18r^−/−^* CTLs was paralleled by a significantly higher average speed (8.05 µm/min) of *Il18r^−/−^* CTLs when compared to WT and *Il1r^−/−^* CTLs (WT: 4.83 µm/min, *Il1r^−/−^*: 4.85 µm/min) in the DSFC model (Figure 1D). Similarly, track displacement (Figure 1E) was significantly higher for *Il18r^−/−^* CTLs when compared to those for WT and *Il1r^−/−^* CTLs. Importantly, all three CTL populations that were transferred into PancOVA tumors had higher track displacement than that of CTLs found in Panc02 tumors, indicating directed, antigen-dependent motility in PancOVA tumors that express the model antigen. The receptor deficiency of approaching and peripheral T cells did not result in marked increases or decreases in average speed (Appendix A).

### 3.2. T-Cellular IL-1R and IL-18R Signaling Inhibits the Rejection of Pancreatic Cancer Spheroids

In preparation of the motility measurements in a pancreatic cancer spheroid model, rejection kinetics were performed. Panc02/PancOVA cells were cultured with Matrigel for spheroid formation and CTLs were added after 48 h. Figure 2A shows representative images of tumor spheroids cocultured with CTLs at early (4 h) and late (18 h) time points, depicting the rejection of PancOVA tumors by WT, *Il18r^−/−^*, and *Il1r^−/−^* CTLs. To quantify the rejection kinetics, spheroids were measured at 4 and 18 h after the beginning of the coculture. After 4 hours of coculture, the tumor volume was set to 100%. In the Panc02 spheroids, there was a relative increase in volume to 125% of the initial volume. In contrast, PancOVA spheroids were reduced in size to 69% of the initial volume at the early phase of the coculture with WT OT-1 CTLs. Coculture with *Il1r^−/−^* and *Il18r^−/−^* CTLs resulted in significantly increased tumor rejection (55% of the initial tumor volume after the application of *Il18r^−/−^* CTLs and 47% after the coculture with *Il1r^−/−^* CTLs; Figure 2B). Again, as in the intravital DSFC model, CTLs were divided into those infiltrating the tumor parenchyma, approaching the parenchyma, and staying in the tumor periphery. The pattern of infiltration was consistent with the data in the DSFC model (Figure 2C). In both the spheroid model and the DSFC model, the parenchymal infiltration of OT-1 CTLs was low for Panc02 tumors. In all three OT-1 CTL subgroups infiltrating the PancOVA tumors, there was a shift from the periphery to the parenchyma. In accordance with rejection kinetics, tumor parenchyma infiltration was highest for the *Il1r^−/−^* CTLs.

### 3.3. IL18R-Deficient CTLs Demonstrate Increased Motility in Tumor Spheroids Compared to WT CTLs

The motility pattern analysis of CTLs in Panc02 and PancOVA spheroids was performed via MPM. For that, data from early (4 h, Appendix A) and late (18 h after the start of the coculture, Appendix A) time points were analyzed. Interestingly, the average speed of WT OT-1 CTLs in Panc02 and PancOVA spheroids decreased from the early to the late time point, as did the average speed of *Il1r^−/−^* OT-1 CTLs in the PancOVA spheroids (dots represent individual tracks, pooled data for all experiments in respective experimental group; Figure 3A). In contrast, *Il18r^−/−^* CTLs in PancOVA spheroids, 18 h after the start of the coculture, showed increased average speed that was indicative of increased motility, corroborating our own data on the increased effector function of IL-18R-deficient CTLs when compared to that of WT CTLs (Lutz et al., Cancer Immunology Research, *in press*). In order to better define the CTL compartment that was most affected by IL-18R deficiency in regards to motility, CTLs were grouped into infiltrating, approaching, and peripheral CTLs, and early and late movement patterns were analyzed. Here, the mean average speed of individual experiments was calculated instead of performing pooled analysis (Figure 3B and Appendix A). At the early time point, the average speed was quite uniformly distributed between receptor-deficient and WT groups in infiltrating, approaching, and peripheral CTLs. In all groups, the average speed at late time points was higher in the periphery than that within the spheroid, indicative of a contact-dependent decrease in motility. In particular, infiltrating *Il18r^−/−^* CTLs at the early time point showed similar velocities as approaching and peripheral *Il18r^−/−^* CTLs (infiltrating: 2.03 µm/min, approaching: 2.56 µm/min, peripheral: 2.00 µm/min). At the late time point after 18 h of culture, however, peripheral *Il18r^−/−^* CTLs were significantly faster (mean: 5.06 µm/min) than infiltrating (mean: 3.21 µm/min) and approaching (mean: 2.99 µm/min) T cells. Relative increase of *Il18r^−/−^* T-cell speed compared to WT T cells was similar for infiltrating and peripheral CTLs (99% increase over WT baseline for infiltrating T cells, and 138% increase over baseline for peripheral T cells). Importantly, receptor-deficiency-associated increase in average CTL speed was found only in spheroid-infiltrating *Il18r^−/−^* T cells (Figure 3B), whereas peripheral speed was also increased by IL-1R deficiency (Appendix A). All this corroborates our own data on the intratumoral dysfunction of T effector cells induced by IL-18R signaling [12]. In summary, IL-18R deficiency led to a T-cell-intrinsic increase in motility. Furthermore, CTL track displacement, which is based on the vectorial distance that T cells travel, was determined in the spheroid model. Infiltrating CTLs showed the shortest vectorial distances in all groups at 4 and 18 h after the start of the coculture. Peripheral CTLs, on the other hand, presented the longest vectorial distances at the late time point. The track displacement of infiltrating WT and *Il1r^−/−^* CTLs decreased over time. In contrast, infiltrating *Il18r^−/−^* CTLs traveled longer distances after 18 h than those they did after 4 h of coculture (Figure 3C and Appendix A). Data on track displacement indicate that movement patterns were directional and mediated by antigen-dependent mechanisms. 

### 3.4. IL-18R Signaling Inhibits Epitope-Specific T-Cell Arrest on Target Cells in Subcutaneous Tumors and Tumor Spheroids

T-cell arrest can be used as a surrogate parameter of the interaction between effector CD8^+^ CTLs and their target cells. As described previously, OT-1 CTLs were adoptively transferred 7 days after tumor inoculation using either the Panc02 or the PancOVA PDAC cell line. T-cell motility was monitored via intravital MPM using DSFCs. The instantaneous velocities of individual OT-1 CTLs were plotted against time (Figure 4A). The duration of intervals during which cells moved slower than 4 μm/min, referred to as pausing phases [12], was calculated (Figure 4C). Whereas no significant differences were observed between WT CTLs in coculture with Panc02 and PancOVA tumors, a significant increase in the duration of pausing phases in PancOVA tumors cocultured with *IL18r^−/−^* and *Il1r^−/−^* OT-1 CTLs was observed. This indicates a more stable engagement of the T cells with the surface antigen. To further investigate the dynamics of these pauses, the arrest coefficient of individual T-cell tracks in different experimental groups was analyzed (Figure 4E). The arrest coefficient is defined as the fraction of time during which each individual cell is immobilized with instantaneous velocity under a certain threshold, which was computed from xyzt data generated by Imaris using a code written in R software. This parameter tends to be lower when T cells are not engaged in stable contacts with their target cells. Results were pooled from three independent experiments for all groups. There was a significant increase in the arrest coefficient of CTLs transferred into PancOVA-bearing animals and visualized in DSFCs as compared to the tumors of the Panc02 control group. This indicates that the presence of the cognate SIINFEKL antigen resulted in antigen-dependent effector contacts with the target cells of the OT-1 CTLs. The arrest coefficient was increased further in *IL18r^−/−^* and *Il1r^−/−^* CTLs, indicating the formation of temporary interactions with the PancOVA tumor cells (Figure 4D).

Having established increased arrest coefficients of *Il18r^−/−^* OT-1 CTLs in PancOVA tumors visualized by DSFCs, raw data on CTL migration in Panc02/PancOVA tumor spheroids were analyzed accordingly. Instantaneous velocities in the spheroid model exhibited similar migration patterns. However, as the velocities of CTLs were lower in spheroids, the threshold for the definition of a pausing phase had to be lowered to 2 µm/min (Figure 4B). Here, again, *Il18r^−/−^* CTLs showed the largest proportion of paused cells (Figure 4F). Again, the arrest coefficient was highest when characterizing *Il18r^−/−^* OT-1 CTLs in PancOVA spheroids (Figure 4H).

### 3.5. IL-18R-Deficient Intratumoral T Cells Exhibit Transcriptional Changes to Genes Involved in Motility

In order to investigate the transcriptional consequences of the IL-18R deficiency of intratumor T cells, RNA-Seq data derived from intratumoral *Il18r^−/−^* versus WT OT-1 CTLs in PancOVA tumors were reanalyzed with a focus on T-cell motility. For this, CD45.1^+^ mice had been subcutaneously inoculated with PancOVA cells. Seven days later, CD45.2^+^ OT-1 CD8^+^ CTLs that were either WT (*n* = 4) or IL-18R-deficient (*n* = 4) were adoptively transferred with an intraperitoneal injection. Three days later, CD45.2^+^ CD8^+^ T cells were sorted on an FACS Aria III, and RNA-Seq was performed. Genes were sorted according to the probability of differential expression. Among the top 10 genes with differential expression in WT versus *Il18r^−/−^* T cells, Coro2a ranked 7th with mean counts of 4942 (WT) versus 1587 (*Il18r^−/−^*) (*p* = 0.0010. Figure 5A). Genes that were up- and downregulated within the RNA-Seq dataset were defined on the basis of thresholds log2FC 0.75 and padj < 0.01. Pathway enrichment analysis was performed using Metascape [26]. Among the top 20 differentially regulated modules, clusters for the negative regulation of cell–cell adhesion and the positive regulation of cell migration were found (Figure 5B). Network plotting revealed a close association of those two gene sets with genes involved in the regulation of cytokine production (ranked 2nd) and T-cell activation (ranked 3rd) (Figure 5C). Heat maps indicate altered gene expression in the locomotion cluster (Figure 5D) and a gene cluster that regulates cell–cell adhesion (Figure 5E).

## 4. Discussion

This paper investigates the role of T cell-intrinsic IL-18- and IL-1-receptor signaling for CTL motility in two models of intratumoral migration. The visualization of migration patterns by intravital MPM in a DSFC model and MPM in a spheroid model found increased average speed of intratumoral *Il18r^−/−^* CTLs. Importantly, arrest coefficients, which are indicative of antigen-mediated cytotoxic effector contacts between T cells and their target cells, were increased in *Il-18r^−/−^* CTLs compared to those in WT T cells. IL-1R deficiency also increased the motility of CTLs, but the effect was less pronounced. RNA-Seq data on IL-18R-deficient intratumoral T cells found the enrichment of gene modules involved in locomotion.

IL-18 and IL-1β are target cytokines of the NLRP3 inflammasome complex. The term “inflammasome” was used first in 2002 by Tschopp et al. when defining protein complexes that mediated the stimulation of inflammatory caspases [28]. NLRP3 activation results in the oligomerization of molecules NLRP3, ASC, and caspase-1, which leads to the cleavage of pro-IL-18 and pro-IL-1β into biologically active forms. NLRP3 activation is tightly regulated. The two-hit hypothesis proposes that the nuclear factor-κB (NF-κB)-mediated transcription and expression of NLRP3 inflammasome components, and pro-IL-1β and pro-IL-18 depends on an initial hit mediated by danger signals such as LPS. These components and proforms remain inactive until a second hit induces the assembly of the inflammasome. As chronic inflammation plays a decisive role in cancer development, NLRP3, IL-18, and IL-1β have been the focus of tumor researchers for over a decade [29]. However, datasets on multiple tumor entities in various models suggest that inflammasomes might play a contrasting role in cancer development and progression [30]. Before compounds such as IL-1β neutralizing antibodies and IL-18 binding protein can be clinically applied, a deeper understanding of the biological mechanisms of NLRP3 activation, IL-18R- and IL-1R signaling, in regards to tumor progression is needed.

T-cell motility is pivotal for T-cell effect function [12]. In the tumor biology, multiple studies found evidence that reduced T-cell motility is associated with increased tumor progression [13,31,32]. MPM imaging has elucidated some important mechanisms, such as the contribution of stromal compartments and collagen fibers, for T-cell migration [32]. On the basis of these data, we decided to investigate T-cell motility in three compartments: parenchyma-infiltrating, approaching, and peripheral T cells. 

Even more importantly, T-cell motility is influenced by blocking coinhibitory molecules that counteract on mechanisms of intratumoral T-cell exhaustion [33,34]. Intravital two-photon laser scanning microscopy showed that CTLA-4 increases T-cell motility and overrides the T-cell receptor (TCR)-induced stop signal required for stable contacts between T cells and APCs. This resulted in reduced contact periods, which in turn decreased effector function. However, the interconnection of increased motility and antitumor effects is regulated on various levels. Following up on CTLA-4 inhibition, Ruocco et al. found that a combination of anti-CTLA-4 therapy and radiation (which induced NKG2D ligands) resulted in the restoration of T-cell stopping and successful tumor treatment [33]. A report by Pentcheva-Hoang et al., however, found that the injection of anti-CTLA-4 antibodies increased the velocities of T cells in tumor-draining lymph nodes [31]. These data encouraged us to investigate T-cell motility in *Il18r^−/−^* and *Il1r^−/−^* intratumoral CTLs, as our own results indicate that T cell-intrinsic signaling through these two receptors induced T-cell exhaustion (Lutz et al., Cancer Immunology Research, *in press*). The data provided here strengthen the notion that intratumoral T-cell exhaustion and alterations to intratumoral T-cell migration are deeply connected.

The mechanistic interplay between T-cell dysfunction and motility was demonstrated by a report that antigen-recognition-induced T-cell stopping was followed by the upregulation of PD-1, which was correlated with regaining motility. The authors concluded that checkpoint inhibition might result in prolonged T-cell arrests, thereby increasing the efficacy of T-cell responses [35]. A causal connection between T-cell exhaustion and T-cell motility is supported by the results from models of chronic viral infection. Zinselmeyer et al. reported on the prolonged motility paralysis of virus-specific CD8^+^ T cells—a phenomenon that was associated with T-cell exhaustion. In this model, the blockade of PD-1–PD-L1 restored CD8^+^ T-cell motility within 30 min. These results were established in a splenic imaging model [36]. Intratumoral T-cell migration might be regulated by quite a different mechanism. In fact, earlier reports found that the cytotoxic contact times of T cells are rather long in tumor immunology when compared to those in viral infection models [37].

We do not suggest that the effects of T-cellular IL-18R- and IL-1R-deficiency on intratumoral motility that we describe here are directly linked to T-cell exhaustion. In order to argue that point, the tumor load would have to be carefully titrated in order for it to reach an equilibrium of tumor rejection and T-cell exhaustion. This is almost impossible to reach in a DSFC model and still hard to calibrate in our PancOVA spheroid model. Instead, we propose that *Il18r^−/−^* (and to a lesser extent *Il1r^−/−^*) CD8^+^ T cells bear intrinsic alterations that render them faster than WT cells (higher average speed), but also increase their efficiency to engage with target cells (higher arrest coefficient). Mechanistically, the expression of Coro2a ranks 7th among differentially expressed genes on RNA-Seq. Coronins are important regulators of the actin cytoskeleton [38], and CORO2A was studied with regard to cellular motility [39]. The overexpression of CORO2A increased tumor cell migration. However, its molecular effect on T-cell motility is unclear. Pathway enrichment analysis supports the notion that genes involved in cell–cell adhesion and the regulation of cell migration are differentially expressed in intratumoral *Il18r^−/−^* versus WT T cells.

The activation of NLRP3, and the subsequent release of IL-18 and IL-1β have effects on multiple cell types, possibly with contradicting effects [40]. The overexpression of NLRP3 in esophageal squamous cell carcinoma cell lines markedly promoted migration and invasion [41]. Interestingly, CD4^+^ T cells needed to be primed by NLRP3 inflammasome-sufficient antigen-presenting cells to upregulate chemotactic proteins such as osteopontin, CCR2, and CXCR6 in experimental autoimmune encephalomyelitis (EAE) [42]. This demonstrated that the NLRP3 inflammasome plays a critical role in inducing T-helper cell migration into the CNS. NLRP3 activation might have differential effects not only on different cell types (such as tumor, myeloid, and T cells), but also on the various stages of the life cycle at which a T cell, for example, might be. Elegant models were described in order to investigate T-cell motility in lymph nodes [43] where antigenic priming takes place. Here, we investigated the mechanisms of the last stages of T-cell cytotoxicity by visualizing the intratumoral migration of those cells. NLRP3-dependent proinflammatory cytokine IL-18 influenced the migration pattern and interaction behavior of intratumoral CTLs. It remains to be investigated how NLRP3 deficiency, and IL-18R and IL-1R signaling affect the earlier phases of T-cell motility, such as the T-cell scanning of APC in lymph nodes, CTL egress, or entry from the blood stream into tumor tissue. An important caveat of this study is the use of only one cell line, Panc02, and its derivate, PancOVA. However, two models were applied, one of intravital imaging using DSFCs and a complex spheroid model, with confirmatory results in both models. Here, our data indicate that the spheroid model was able to recapitulate characteristics of adoptive T-cell transfer into tumor-bearing mice, acting as an eligible and versatile model system for the reduction in, and refinement and replacement (3R principle) of research on intratumoral T cells. However, a spheroid model is always limited in immunological complexity. We plan to further investigate rejection kinetics and cell–cell interactions within the tumor spheroid with nonimaging techniques in the future. Then, combining these results with imaging data from the spheroid model, and the more complex and physiologically relevant intravital dorsal skinfold chamber model would gain further insights into the causal relationship of T-cell plasticity, in particular T-cell exhaustion, and intratumoral migration.

## Figures and Tables

**Figure 1 cells-12-00456-f001:**
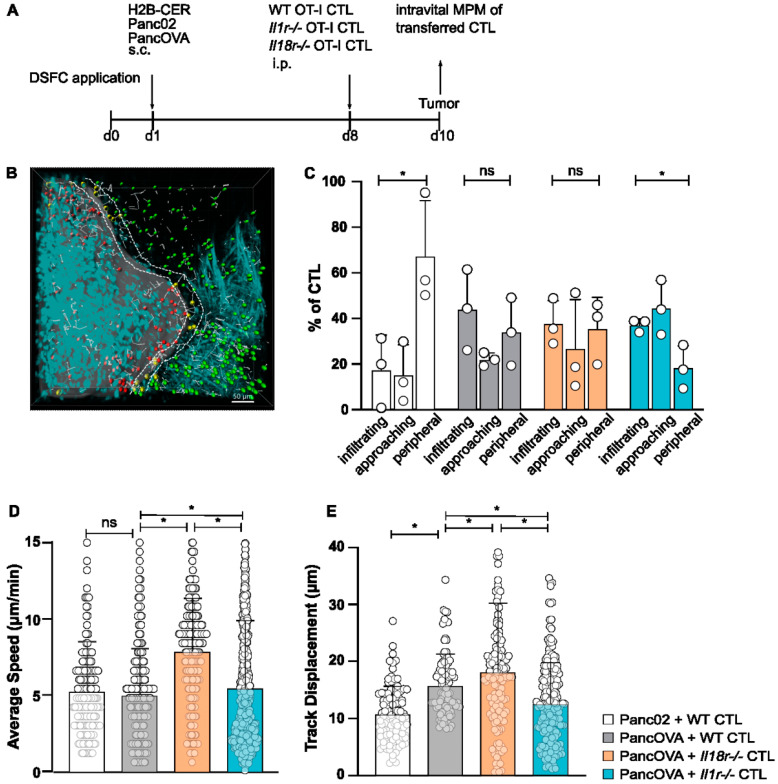
**IL-18R deficiency enhances in vivo intratumoral CTL migration**. (**A**) Schematic representation of the experimental design. After the surgical implantation of dorsal skinfold chambers (DSFCs), Panc02/PancOVA H2B cerulean tumor cells were inoculated into the imaging chamber. After 7 days, CTLs generated from OT-1 wild-type mice, *Il1r^−/−^* OT-1 mice or *Il18r^−/−^* OT-1 mice were adoptively transferred. Imaging via multiphoton microcopy (MPM) was performed 36 h after transfer. (**B**) Representative three-dimensional reconstruction of a region with a Panc02 tumor in the DSFC and adoptively transferred WT CTLs. Tumor cells are colored turquoise. CTLs are colored depending on their position in relation to the tumor: in the tumor (infiltrating, red), close to the tumor (approaching, yellow), and far away from the tumor (peripheral, green). The motility of T cells is depicted as gray lines (scale bar = 50 µm). (**C**) The antigen-specific infiltration of OT-1 CTLs into subcutaneous PancOVA tumors as visualized with intravital MPM. The number of CTLs was determined via Imaris software with a spot algorithm after separation into different regions: inside the tumor (infiltrating), <10 µm to surface of tumor (approaching), and >10 µm to surface of tumor (peripheral). Results are expressed as the mean ± standard deviation (SD) for *n* = 3 biological replicates per group; an unpaired, two-tailed *t*-test was performed; * *p* < 0.05. (**D**,**E**) The migratory pattern of infiltrating WT, *Il1r^−/−^*, and *Il18r^−/−^* CTLs was analyzed with a spot algorithm in Imaris software via 30 min DSFC videos. (**D**) The average speed of tumor-infiltrating IL-18R-deficient versus WT and IL-1R-deficient OT-1 CTLs in the DSFC, as determined with intravital MPM. Each dot represents one individual tracked T cell. Graph represents the pooled data from *n* = 3 experiments, with 205–529 cells per subgroup. Statistical analysis was performed with the Kolmogorov–Smirnov test; * *p* < 0.05. ns = not significant. (**E**) Track displacement indicating the directed motility of WT, *Il1r^−/−^*, and *Il18r^−/−^* CTLs in Panc02/PancOVA tumors in the DSFC. Each dot represents one individual tracked T cell. Graph represents the pooled data from *n* = 3 biological replicates per group, with 297–470 cells per row. Statistical analysis was performed with the Kolmogorov–Smirnov test; * *p* < 0.05.

**Figure 2 cells-12-00456-f002:**
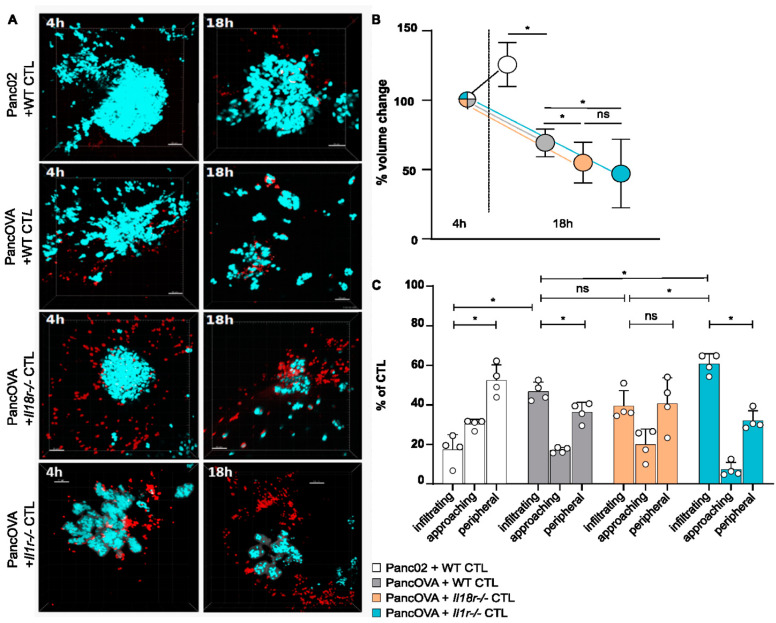
**T-cellular IL-1R and IL-18R signaling inhibits the rejection of pancreatic cancer spheroids**. (**A**) Representative three-dimensional reconstructions of H2b Cerulean Panc02 or H2b Cerulean PancOVA tumor spheroids in coculture with OT-1 CTLs after 4 and 18 h. Tumor cell spheroids were generated by using Corning Matrigel. CTLs were derived from WT, *Il1r^−/−^*, or *Il18r^−/−^* OT-1 mice that expressed a transgenic T-cell receptor recognizing OVA-derived peptide SIINFEKL. Tumor cells are depicted in turquoise, CTLs in red; scale bar = 50 µm. (**B**) Imaris software was used to calculate the spheroid volume of 5 randomly chosen spheroids from MPM-derived z-stacks. Initial tumor volume after 4 h of coculture was normalized to 100%. The relative growth/regression of spheroids cocultured for 18 h with WT, *Il1r^−/−^*, and *Il18r^−/−^* CTLs was calculated. Graph represents the results of *n* = 6–13 independent experiments per subgroup. ns = not significant. (**C**) The antigen-specific infiltration of OT-1 CTLs into PancOVA spheroids via intravital MPM. The number of CTLs was detected through analysis via Imaris software with a spot algorithm after separation into different regions: inside the spheroid (infiltrating), <10 µm to surface of the spheroid (approaching), and >10 µm to the surface of the spheroid (peripheral). Results are expressed as the mean ± standard deviation (SD) for *n* = 4 independent experiments per group. An unpaired, two-tailed *t*-test was performed; * *p* < 0.05. ns = not significant.

**Figure 3 cells-12-00456-f003:**
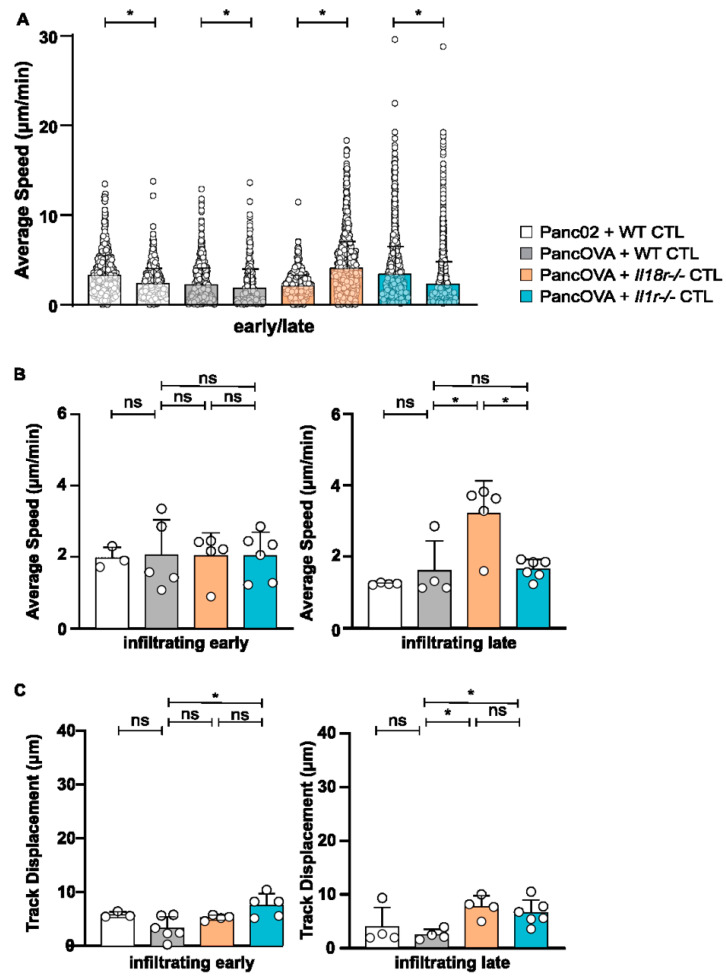
**IL18R-deficient CTLs demonstrated increased motility in tumor spheroids compared to that of WT CTLs**. The migratory patterns of WT, *Il1r^−/−^*, and *Il18r^−/−^* CTLs were analyzed with a spot algorithm in Imaris software via 30 min movies of a Panc02/PancOVA tumor spheroid coculture at early (4 h after the start of the T-cell coculture) and late (18 h) time points. (**A**) The average speed of all WT, *Il1r^−/−^*, and *Il18r^−/−^* CTLs per experiment at the early and late time points. Each dot represents one individual tracked T cell. Graph represents pooled data from *n* = 3–6 experiments with 431–2761 cells per subgroup. Statistical analysis was performed with the Kolmogorov–Smirnov test; * *p* < 0.05. (**B**) The average speed of infiltrating WT, *Il1r^−/−^*, and *Il18r^−/−^* CTLs at the early and late time points. The mean of every experiment (*n* = 3–6) is shown as a dot. An unpaired, two-tailed *t*-test was performed; * *p* < 0.05. ns = not significant (**C**) Track displacement indicating the directed motility of infiltrating WT, *Il1r^−/−^*, and *Il18r^−/−^* CTLs at the early and late time points. The mean of every experiment (*n* = 3–6) is shown as a dot. An unpaired, two-tailed *t*-test was performed; * *p* < 0.05. ns = not significant.

**Figure 4 cells-12-00456-f004:**
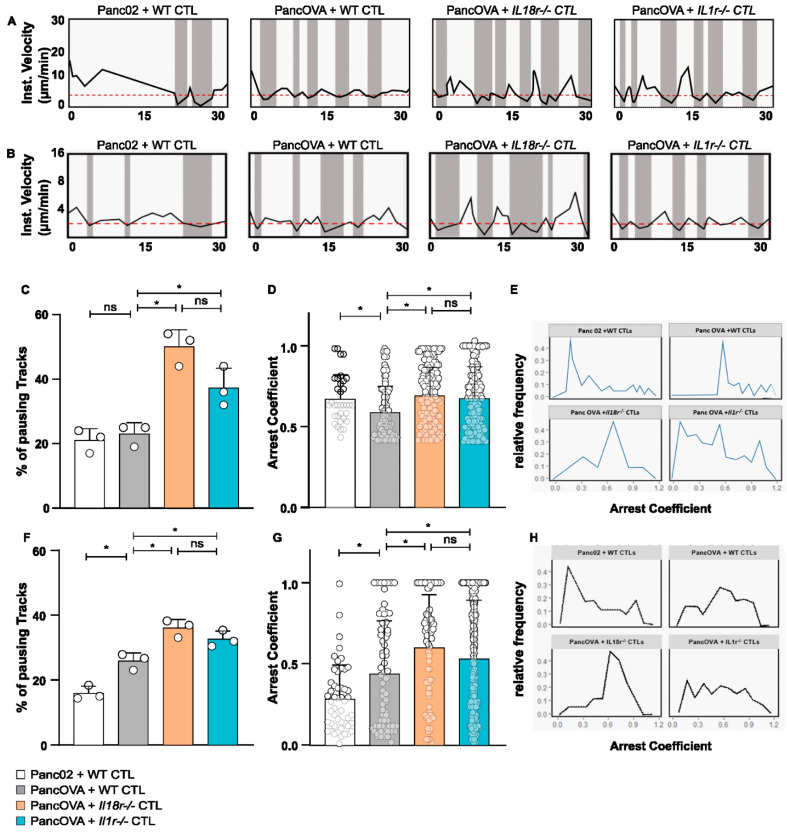
**IL-18R signaling inhibits epitope-specific T-cell arrest on target cells in subcutaneous tumors and tumor spheroids**. (**A**) Representative graph of the instantaneous velocity of individual tracked WT, *Il1r^−/−^*, and *Il18r^−/−^* CTLs in subcutaneous DSFC tumors. The graphs show the alteration of high and low instantaneous velocities. An instantaneous-velocity threshold of 4 µm/min (dashed red line) was set to mark the pausing phases. (**B**) Representative graph of the instantaneous velocity of individual tracked WT, *Il1r^−/−^*, and *Il18r^−/−^* CTLs in co-culture with Panc02/PancOVA tumor spheroids. The graphs show the alteration of high and low instantaneous velocities. The threshold of instantaneous velocity (dashed red line) representing pausing phases had to be changed to 2 µm/min, as CTLs in the Matrigel spheroids generally moved slower than T cells in the DSFC tumors. (**C**) Pausing phases of WT, *Il1r^−/−^*, and *Il18r^−/−^* CTLs in subcutaneous DSFC tumors as visualized by intravital MPM. The graph represents the percentage of CTLs moving with instantaneous velocity < 4 µm/min. Individual dots represent the pooled data of 3 biological replicates. Statistical analysis was performed via an unpaired, two-tailed *t*-test; * *p* < 0.05. ns = not significant (**D**) The arrest coefficient of WT, *Il1r^−/−^*, and *Il18r^−/−^* CTLs in the subcutaneous tumor in the DSFC is depicted. The arrest coefficient is defined as the fraction of time during which an individual cell has instantaneous velocity < 4 µm/min, which is indicative of an interaction with tumor cells. The graph represents pooled data from *n* = 3 biological replicates. Statistical analysis was performed with the Kolmogorov–Smirnov test; * *p* < 0.05. The arrest coefficient was calculated with R Studio. For visual clarity, only data points >0.4 are shown. ns = not significant (**E**) Relative frequency of arrest coefficient is depicted, representing pooled data from *n* = 3 biological replicates. Analysis was performed by calculating the arrest coefficient for different experimental groups (WT CTLs in Panc02 and PancOVA tumors, *Il1r^−/−^* and *Il18r^−/−^* CTLs in PancOVA tumors) on the basis of imaging data from intravital MPM in DSFC tumors. (**F**) Pausing phases of WT, *Il1r^−/−^*, and *Il18r^−/−^* CTLs in Panc02/PancOVA tumor spheroids. Graph represents the percentage of CTLs moving with instantaneous velocity < 2 µm/min. Individual dots represent the pooled data of 3 individual experiments. Statistical analysis was performed with an unpaired, two-tailed *t*-test; * *p* < 0.05. The arrest coefficient was calculated with R Studio. ns = not significant (**G**) The arrest coefficient of WT, *Il1r^−/−^*, and *Il18r^−/−^* CTLs in Panc02/PancOVA tumor spheroids is depicted. The arrest coefficient is defined as the fraction of time during which an individual cell has instantaneous velocity < 2 µm/min, which indicates an interaction with tumor cells. The graph represents the pooled data from *n* = 3 individual experiments. Statistical analysis was performed with the Kolmogorov–Smirnov test; * *p* < 0.05. The arrest coefficient was calculated with R Studio. ns = not significant (**H**) The relative frequency of the arrest coefficient is depicted in the graph, representing pooled data from *n* = 3 biological replicas by calculating the arrest coefficient for the different experimental groups (WT, *Il1r^−/−^*, and *Il18r^−/−^* CTL) on the basis of imaging data from Panc02/PancOVA tumor spheroids.

**Figure 5 cells-12-00456-f005:**
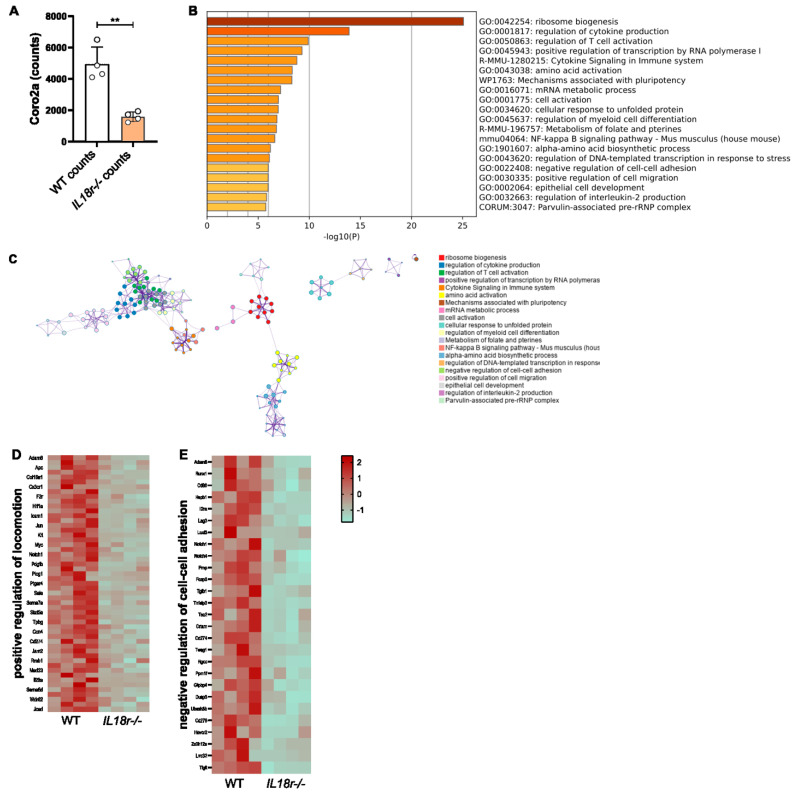
**IL-18R signaling alters the transcription profile of motility-associated genes expressed in intratumoral CTLs.** (**A**) RNA-Seq data were analyzed for the differential expression of genes that were ranked according to padj values. Coro2A expression ranked 7th among all genes with differential expression, ** *p* < 0.01. (**B**) The histogram of clustered enriched terms across input gene lists, color-coded on the basis of *p*-values. (**C**) Network of enriched terms; each color represents a distinct cluster ID. Nodes standing close to each other share the same cluster ID. (**D**,**E**) Heatmaps representing the Z-score visualization of the RNA-Seq results for (**D**) the positive regulation of locomotion and (**E**) negative regulation of cell–cell adhesion. Protein-coding genes that show significantly differential expression in WT and *Il18r^−/−^* T cells are depicted.

## Data Availability

Not applicable.

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
