# Peer review of "IL18 Receptor Signaling Inhibits Intratumoral CD8+ T-Cell Migration in a Murine Pancreatic Cancer Model"

_cells, 2023, doi:10.3390/cells12030456_

Round 1
Reviewer 1 Report
The authors of the paper entitled “IL18 receptor signaling inhibits intratumoral CD8+ T cell migration in a murine pancreatic cancer model” focused on the contribution of IL-18R signalling to T cell 34 effector strength, warranting further investigation on phenomena such as intratumoral 35 T cell exhaustion. The presented manuscript is unique as this area is not yet explored. I have a couple of insights on paper.
Materials and Methods:
1. MICE:
Please indicate the sex and age of the animals. In what conditions were the animals bred (line 115)?
2. MURINE T CELL ISOLATION AND IN VITRO DIFFERENTIATION:
Were the animals anaesthetized (line 118)?
Was an antimicrobial agent added to the cell medium (lines 121-123)?
On what basis was a such time (5 days) selected? Literature data evaluation or experimental selection (line 123)?
3. STAINING OF CYTOTOXIC T CELLS
Add the information that the dye is transferred to daughter cells, but not adjacent cells in the population.
Reviewer 2 Report
1. A very valuable and high-quality study has been done, so it is better to clearly state the gap in the introduction
2. Why did you use the times 4h and 18h in your study?
3. It is better to express the limitations of your study and suggestions for future research in the discussion.
4. Usually, except in special cases, references to other researches are not given in the results section, and it is recommended that you only express your findings in the results section.
It is not necessary to mention a very old research related to 1984
5. Try to use the same style of writing in all the text of the article (Fig. , fig. ….)
6. The keywords of this manuscript are not well written and it is necessary to use appropriate keywords.
Reviewer 3 Report
Summary:
E. Nasiri et al. utilized multiphoton microscopy in a dorsal skinfold chambers model and in a spheroid model to investigate the role of T cell-intrinsic IL-1- and IL-18-receptor signaling for CD8+ cytotoxic T cell motility in the pancreatic tumor. Visualization of migration patterns and measurement of migration speed, tumor volume, as well as T cell arrest coefficients show that destruction of T-cellular IL-1- and IL-18-receptor signaling enhances T cell motility and T cell arrest, leading to reduced tumor volume of pancreatic cancer. Their data indicate the translational significance of T-cellular IL-1- and IL-18-receptor signaling in cytotoxic effector function.
General concept comments:
1. This manuscript lacks complementary experimental data for most of the conclusions. To target journals like Cells, investigation of molecular mechanisms can be necessary.
Specific comments:
1. Figure 1B is a representative intravital multiphoton image of the infiltrating, approaching, and peripheral Il1r-/- CTL with a PancOVA tumor, showing the largest proportion of peripheral and the least proportion of approaching T cells, which is inconsistent with the statistical data of PancOVA + Il1r-/- CTL combination in Figure 1C.
2. In Line 226 and Line 227, the authors claim that significant infiltration of CTL into tumor parenchyma from the periphery was only found for Il18r-/- CTL, while Figure 1C shows that PancOVA + WT CTL combination has the highest percentage of infiltrating CTL. The description is inconsistent with the statistical data in Figure 1C.
3. For the assessment of rejection kinetics in pancreatic cancer spheroids, other complimentary analyses should be conducted to explore the functional consequences of T cell infiltration and activation in spheroids (e.g. global activation of caspase-3 and -7 in spheroids by live-imaging, quantification of AnnexinV+DAPI+ apoptotic tumor cells by flow cytometry).
Round 2
Reviewer 3 Report
All of my comments have been addressed. I believe the manuscript has been sufficiently improved to warrant publication in Cells.
Author Response
We thank the reviewer for the valuable comments that helped us to improve the manuscript.